# Measuring Māori Health, Wellbeing, and Disability in Aotearoa Using a Web-Based Survey Methodology

**DOI:** 10.3390/ijerph20186797

**Published:** 2023-09-21

**Authors:** Tristram R. Ingham, Bernadette Jones, Meredith Perry, Martin von Randow, Barry Milne, Paula T. King, Linda W. Nikora, Andrew Sporle

**Affiliations:** 1Department of Medicine, University of Otago, Wellington 6242, New Zealand; bernadette.jones@otago.ac.nz; 2Foundation for Equity and Research New Zealand, Wellington 6147, New Zealand; 3Te Ao Mārama Aotearoa Trust, Wellington 6037, New Zealand; 4School of Physiotherapy, University of Otago, Dunedin 9016, New Zealand; meredith.perry@otago.ac.nz; 5Compass Research Centre, University of Auckland, Auckland 1142, New Zealand; m.vonrandow@auckland.ac.nz (M.v.R.); b.milne@auckland.ac.nz (B.M.); 6Te Rōpū Rangahau Hauora a Eru Pōmare, University of Otago, Wellington 6242, New Zealand; paula.king@otago.ac.nz; 7Nga Pae o te Māramatanga, Faculty of Arts, University of Auckland, Auckland 1142, New Zealand; l.nikora@auckland.ac.nz; 8iNZight Analytics Ltd., Auckland 1010, New Zealand; 9Department of Statistics, Faculty of Science, University of Auckland, Auckland 1142, New Zealand

**Keywords:** indigenous methodology, Kaupapa Māori research methods, indigenous survey design, disability

## Abstract

High-quality evidence on the prevalence and impact of health, wellbeing, and disability among Māori, and other Indigenous peoples, is crucial for mitigating health inequities. Current surveys are predominantly centred within a biomedical paradigm, with the constructs mismatched with Indigenous worldviews. We aimed to develop and deploy an accessible and culturally grounded survey exploring Māori health, wellbeing, and disability using a Kaupapa Māori Research methodology. An extensive codesign process with Māori community partners interrogated all aspects of the design to ensure the process and outcomes met the needs of Māori. A large-scale, nationally representative survey of people of Māori descent was conducted. We used a multi-modal deployment approach that included online and alternate methods of completion. Our analysis included a novel dual-weighting system to ensure generalisability of results to the national Māori population. This achieved a survey of 7230 participants, a sample size comparable with government-administered surveys. The response rate was 11.1%, with 7.3% opting for alternate methods. A high completion rate of 93.4% was observed. This approach demonstrated a high level of engagement, resulting in an unprecedented collection of Māori health, wellbeing, and disability data. This highlights the importance of Indigenous codesign for ensuring accessible and culturally appropriate survey methods.

## 1. Introduction

Disability is a global issue, with about 15 percent of the world’s population living with some form of disability and the prevalence is rising due to an aging population and the rapid increase of chronic conditions [1]. Disability has become increasingly accepted as a human rights issue following the United Nations Convention on the Rights of Persons with Disabilities (UNCRPD). Disability is also an important development issue, with an increasing body of evidence showing that people with lived experience of disability encounter worse socioeconomic outcomes and poverty than people without lived experience of disability.

Along with enduring discrimination based on disablism, Indigenous people with lived experience of disability are even more likely to experience the impacts of other forms of oppression, including discrimination arising from racism [2,3]. Additionally, Indigenous people worldwide have diverse historical and contemporary effects of disablement arising from the impacts of colonisation that are in themselves disabling [4]. Māori, as the Indigenous people of Aotearoa New Zealand (NZ), claim a unique partnership relationship legislated under Te Tiriti o Waitangi (the Māori version of the Treaty of Waitangi) as tāngata whenua (people of the land)—see Appendix A (Table A1). Māori continue to suffer inequitable health outcomes due to colonial Western models of government, and there is growing evidence towards an association between disability, multidimensional poverty, and inequities [5].

While Māori initiatives have challenged contributors to these inequities, such as institutional racism, there remains a scarcity of evidence and initiatives related to Māori and disability. Existing challenges for Māori are that the dominant medical and social models, disability measures, and related data collection frameworks do not account for the social, cultural, and environmental context of Māori. Lack of definitional clarity and robust statistics, along with little attention to, or funding for, disability research, has resulted in a limited evidence base and evidence gaps in all sectors [6]. Few data are available that identify the evidence from the perspective of Indigenous peoples, such as Māori, or that recognise the impact of racism and colonisation on the additive impacts for these groups [4]. Even within te ao Māori (the Māori world) and between communities, viewpoints on disability and disablement occur. As with other measures of Māori health and wellbeing, Māori disability data are living taonga (treasure) and measures should be developed and governed by Māori to advance the aspiration of tāngata whaikaha Māori (Māori with lived experience of disability) [7].

For a range of reasons, current survey methods with methodologies and question sets originating from the Global North do not resonate effectively with Māori and other Indigenous peoples, leading to lower engagement and response rates. These reasons include denominator bias, self-identification, and interpretation by non-Indigenous analysts [8]. There is a need to better understand the performance and sociocultural contextualisation of the disability items in the current census to more accurately inform future national data collection methods. Our broader research project aimed to extend an understanding of Māori cultural perspectives of disability identity by ensuring that national survey questions accurately quantify the prevalence of disability among Māori and to use these prevalence estimates to quantify the impacts of disability [5].

This paper describes our approach and methods in which we endeavour to uphold te ao Māori principles as we work in partnership with tāngata whaikaha Māori to implement robust yet culturally appropriate research processes. We have used the Checklist for Reporting Results of Internet E-Surveys (CHERRIES) to report our processes [9].

## 2. Methodology

The approach underpinning the design of this nationally representative survey was grounded in a Kaupapa Māori Research methodology. A Kaupapa Māori Research (KMR) paradigm is an emancipatory, Indigenous research paradigm connected to Māori philosophy and principles [10,11,12]. KMR takes for granted the validity and legitimacy of Māori and the importance of Māori language and culture, with a focus on autonomy over Māori health and wellbeing [13]. KMR is concerned with both the methodological developments and the forms of research methods utilised. In this sense, KMR is described as both a theory and an analysis of the context of research involving Māori, with the approaches to research expressed as being by, with, and for Māori [10].

To align with this methodological approach, an extensive process was undertaken by the investigators to ensure critically interrogating of all aspects of the research methods to ensure they remained tika (correct) and pono (genuine), with a te ao Māori world view [14], and met tāngata whaikaha Māori needs and aspirations [15]. We collectively ensured participation was mana enhancing (respectful), followed tikanga (Māori values), and benefited the people and communities who were involved in the research at every stage of the project from initial conception to dissemination [16].

The concept for this research had been initiated by the Māori disability community, and cultural consultation had occurred with kaumātua (expert Māori elders). To formalise this process, prior to designing the research project, securing funding, and developing the survey, a partnership was developed with an existing nationally representative tāngata whaikaha Māori group and their local regional rōpū. The advisory group consisted of fifteen tāngata whaikaha Māori with expertise in the areas of Māori disability, policy, disability services, tikanga Māori (Māori protocols), and cultural identity. Engagement and advice from the group occurred from the beginning and continued throughout the whole process. Additionally, within an ethical paradigm and in keeping with KMR best practice, “Te Ara Tika Guidelines for Māori Research Ethics” were used to develop and conduct all aspects of this research process [17].

### 2.1. Development and Testing

This survey aimed to estimate the actual prevalence of disability within the Māori adult population. Prior to the deployment of the survey, a multi-stage mixed-methods process was used to develop the survey instrument, as outlined below in Figure 1. Qualitative methods were used to inform our understanding of Māori ‘disability’ identity. The development of the questions (Phase 1) for this survey used a qualitative approach to data collection, incorporating flexible KMR methods. Wānanga (workshops), hui (meetings), and in-depth individual and whānau (extended family) interviews were undertaken to align with Māori ways of gathering information.

#### 2.1.1. Identification of Questionnaire Domains and Priorities

Semi-structured interview guides were used to explore topics within three broad domains: (1) Culture and identity (including disability concepts); (2) health and disability services (including access and discrimination); and (3) transformation of the disability system (suggestions for improvements). The interview guide was developed in collaboration with the kaumātua steering group.

In keeping with KMR, a strengths-based approach ensured all those involved with the project had control over who they shared their stories with and how they wanted them to be interpreted and represented. Further details on the data gathering and analysis can be found in the previously published papers referenced below. In summary, these results emphasise how Western-centric constructs of ‘disability’ fail to align with te ao Māori perspectives of disability, resulting in multidimensional impacts for tāngata whaikaha Māori [5]. The interpretations were inclusive of the notion of “karanga rua, karanga maha” (two identities, or multiple identities) as a potential framework to understand how tāngata whaikaha Māori conceptualise and express a plurality of identities within Māori collectives [18]. Māori ways of being, knowing, relating, and doing are critical to advancing understanding of the impacts of disability and addressing priorities and aspirations of Māori with lived experience of disability.

Whakawhiti kōrero (a traditional process of discussion and negotiation) was used to validate data interpretations and ensure analytic rigour and expert checking. The analysis was discussed through wānanga (the process of knowledge creation) via two processes. Māori researchers held wānanga with participants to confirm the interpretation of their individual data. The researchers held a series of wānanga, led by tāngata whaikaha Māori participants and including the steering group, to facilitate an independent Māori lens in the analysis. This iterative process took place over several months, allowing for a uniquely Indigenous Māori perspective on disability that is holistic and based on spiritual, collective, and relational values.

#### 2.1.2. Survey Question Development and Compilation

Having identified Māori priorities for disability and wellbeing, we then reviewed the literature on validated national and international instruments, prioritising literature relating to Indigenous measures. A mix of validated and new questions was then used to ensure a holistic approach to the development of the questionnaire. Refer to Appendix A (Table A2) for a high-level overview of the topic domains that were included in the survey, along with the provenance of specific validated question sets. Previously validated questions were drawn from a range of surveys so comparisons could be made with current government data and our survey results. Newly developed questions were generated where gaps existed, particularly incorporating Māori concepts of disability.

This led to the potential inclusion of >500 items. The research team then had multiple meetings to refine which items would be included based on the priorities identified by the semi-structured interviews. Where validated items did not exist, questions were developed by the research team. For example, the preference for disability-related terminology, including kupu Māori (Māori terms). Whakawhiti kōrero was again undertaken over several iterations with the steering group for the sense making of the newly developed items and the sections in which they were placed—see Appendix A (Table A3).

#### 2.1.3. Sector Consultation and Review

Further consultation was held with stakeholders, specifically Statistics New Zealand (Stats NZ), to ensure the survey items would be compatible with other nationally administered surveys and that they would enhance current data collection. Stats NZ provided a peer review of the draft survey and provided recommendations that were incorporated into the final survey. We also reached an agreement that, once validated, new items would inform future national disability surveys. In addition, engagement with the Disability Data and Evidence Work Group (a national cross-government expert data group consisting of government officials, university representatives, and community experts) was undertaken. There was a positive response, with no specific recommendation for survey change and a desire to be kept informed.

#### 2.1.4. Survey Design and Formatting

We recognised the importance of the survey being appropriate, empathetic, relatable, and accessible to the needs of the target audience to balance the often-impersonal nature of surveys that can dissuade responses [19]. It also needed to be culturally safe and relatable to Māori [19]. As part of our collaboration with the steering group during the creation of the qualitative interview schedule, questions were developed about design and ways to increase the acceptability of the survey. Participants responded with specific suggestions regarding features of the survey, visual elements, tone, and progression through the sections, to ensure the survey was identifiable as being authentically created by Māori, with and for Māori.

#### 2.1.5. Cultural Compatibility

Our steering group consultation and advice reinforced the need for the “look and feel” of the survey to be compatible with te ao Māori, to maintain the engagement of survey respondents, and to minimise dropout. Consequently, we engaged a tikanga Māori designer to create visual representations of Māori whānau (extended and encompassing concept of family), whenua (connection to the natural environment) land, and wairua (spiritual connection). Māori symbology, such as tāniko, is based on traditional weaving patterns featured as borders. Branding expertise was commissioned, resulting in the addition of a series of pause breaks between each section of the questionnaire containing a page with Māori imagery, and whakataukī chosen by a tāngata whaikaha Māori kaumātua member of our steering group. The introduction of the survey was immediately recognisable as Māori because of whakataukī (Māori proverbs), and these features were consistently applied across all elements of engagement, starting with the survey branding of the invitation letter and envelope, the study name, domain name, website landing page, and section pause points.

Te reo Māori (the Māori language) translation of the survey questions was undertaken, along with an independent te reo Māori expert who quality checked all translations for validity and sense checking. The letter of invitation was also translated into te reo Māori and both English and te reo Māori versions were sent to those on the Māori electoral roll (n = 36,212), with the English version sent to those on the general electoral roll (n = 33,943). All survey invitations were sent out in custom-designed envelopes with survey branding reflecting the study title—“te ao Mārama” (the world of enlightenment). A branded website at https://teaomarama.maori.nz was established on 13 June 2022. The website served as the landing page for the survey and prominently displayed the link to start the survey. The webpage also contained additional information about the investigators, the ethics approval, and contact details for the team. The dedicated email address received more than 200 messages over the duration of the survey.

To ensure accessibility provision for a range of participants, we also provided the option to (i) complete the survey with an interviewer by phone using an 0800 number; and (ii) use of a paper version of the survey that was sent to participants upon request. The survey format contains skips and conditional questions, which made it harder to navigate with the 20-page paper version which was sent with a prepaid return envelope. A printed version of the questionnaire translated into te reo Māori was available on request.

#### 2.1.6. Accessibility and Language

Following advice from the tāngata whaikaha Māori steering group, the participant information sheet, consent forms, and letters of invitation were written in English using a plain language approach. They were then translated into te reo Māori by expert translators. For those who opted to complete the survey online, the landing page used browser features to enable screen readers, variable font size, and contrast settings. Braille formats and Audio Computer-Assisted Self-Interview (ACASI) software (https://acasillc.com/home.htm) were also considered to facilitate self-completion of the survey by people who are blind or have low vision. We were advised, however, by Māori disability groups to offer a telephone helpline with an interviewer-administered questionnaire as an alternative to the self-completed online option for all participants. The survey was also available in hard-copy postal format for those who preferred this option. Participants could elect whether to view the website or complete the survey (in all three response formats) in either English or te reo Māori.

### 2.2. Sampling Framework and Sample Size

Our sampling framework was NZ citizens or residents on the NZ electoral roll (both the general and Māori rolls) who were aged 18 years and older, identified as being of Māori descent in December 2021, and were residents in NZ (N = 517,909). It is a legal requirement for all New Zealanders aged 18 years and over to be listed on the electoral roll. The Electoral Commission estimates that 89% of eligible voters are enrolled [20]. New Zealanders of Māori descent can opt to be on the general electoral roll or the Māori electoral roll. The electoral roll contains information on name, address, date of birth, occupation, Māori descent, and region. We excluded overseas mailing addresses and used the random sampling function of SPSS (SPSS Statistics v29, IBM Corporation, Armonk, NY, USA) to select a simple random sample of the remaining individuals.

We powered the study based on our primary objective, which was to identify the prevalence of disability among Māori. We sought to obtain 2000 tāngata whaikaha Māori. The sample size was calculated based on achieving an overall margin of error of 2.2% (95% CI). This would also result in an estimated margin of error of 4.2% (95% CI) for each of the four main impairment categories (hearing, physical, psychological, and vision) used by the 2013 New Zealand Disability Survey [21]. Given that 24% of New Zealanders self-reported as disabled in the 2013 New Zealand Disability Survey, we aimed to have 8000 respondents complete the survey. Based on previous sampling for surveys, we recognised the need to invite sufficient people to account for drop-off rates at different points in the process. Accordingly, we allowed for 99% valid addresses, a 14% response rate, and an 80% survey completion rate, thus requiring an anticipated sample size of 70,000 invitations.

### 2.3. Recruitment Process

A personalised letter of invitation explained the purpose of the survey, who was conducting it, the ethical approval, how their names and addresses were obtained, that we had taken a random sample from the electoral rolls, how long the survey would take to complete, that their participation was voluntary, that they could complete the survey online via Qualtrics software (Qualtrics XM, Qualtrics, Seattle, WA, USA), or other options such as by phone with a researcher administering the survey or by a paper copy, and storage of their data. They could also choose to enter a draw for one of ten NZD 100 gift cards as an acknowledgement of their participation. People on the Māori roll were sent two copies of the invitation letter, one in English and the other in te reo Māori. People on the General Roll were sent the invitation letter in English with bilingual headings. People interested in the research were encouraged to go to the survey web landing page where they could find out more about the study and enrol.

The mailout was undertaken in two tranches to allow for the exploration of the initial participation response rate in this specific population. This also allowed for a staggered response to enable the survey administration team and call centre to provide timely support. As the participant response rate distribution was deemed acceptable, both tranches were run from the same sample frame, and no over-sampling was required. Two postcard reminders were sent at three-week intervals, following the initial mailout, to all those who had not yet responded.

### 2.4. Survey Administration

#### 2.4.1. Survey Enrolment and Consent Process

Our survey website was specifically created for the survey and no other purpose, and only participants selected from the electoral rolls were given the study URL. The landing page described the project team and included FAQs about the survey, a study-specific email address for questions about the survey, and a link to begin the survey via Qualtrics. The survey website was made available in Te Reo and English via a toggle switch and was designed to web standards as per the Web Content Accessibility Guidelines (WCAG 2.1) [22].

As a reasonable accommodation, for participants who needed a personal approach or may have limited internet accessibility, a survey free-phone number was available for all participants between the hours of 9 am and 9 pm on weekdays. Monitoring of the call line was conducted by a dedicated inbound–outbound contact centre with extensive experience in survey administration, with all staff trained and supervised by the research team. The team answered questions about the survey and directly entered respondent data into the Qualtrics survey on behalf of participants. A paper-based version of the web survey was also available for people who explicitly requested this format.

#### 2.4.2. Confirmation of Consent

Having been fully informed about the survey and the opportunity for all questions to be answered, people entered the survey via a link into Qualtrics. On entering the Qualtrics survey, participants were informed that proceeding to complete the survey was taken as consent and were presented with a click button to confirm consent, commence the survey, and complete the demographic verification fields. Consent was confirmed verbally for those completing the survey via the free phone number. The return of completed paper surveys was considered obtained consent.

### 2.5. Survey Format

The first item asked participants to enter their unique identification number (ID) from their letter of invitation. Each ID was a randomly generated 5-digit number used primarily to confirm the identity of survey participants and to ensure that the survey was open to invited participants only. Participants were given the option to complete the survey in te reo Māori or English using the multiple language function of Qualtrics, which automatically aggregates both versions into the same data set. The order of survey items was standardised. Adaptive questioning was used so that only relevant items were displayed and to build on the logic of previous responses. This minimised participant burden and gave the sense that the survey was more personalised.

There were 5 sections with 25 screens. Items included a combination of categorical, ordinal (Likert scales), and open response data. A progress bar showed percentage completeness at the top of each screen. The number of items per page varied due to adaptive questioning. The minimum number of items needed to complete the survey was 97, and the maximum number was 153. The paper version of the survey was presented on 20 pages, again with variable numbers of items per page, and all adaptive questioning paths were shown. Participants were able to review and change their answers using a “Back” button, which displayed previous responses. Completeness checks were performed after survey submission, and duplicates were checked for as part of the process. The survey ID question was the only enforced answer, with non-response options available for all remaining items.

### 2.6. Survey Validation

Unique site visitors were not tracked due to the previously mentioned undercounting that would result from the use of shared IP addresses from VPN services. Accordingly, the view rate was also not calculated. Participants were able to complete the survey over several sessions. Cookies were set and read by Qualtrics to enable the progression of subsequent sessions. Cookies prevented multiple completions by a single user, provided they were on the same computer with the same browser, and enabled Qualtrics to keep track of when an incomplete survey should be recorded as a response. Cookies allowed each survey response to remain open and valid for a maximum of 1 month.

Survey ID number and address were used to identify duplicates and to cross-check against the electoral roll data extract to verify identity. In rare cases where a duplicate response was submitted, the first complete response was retained. If participants completed different sections of the survey in different submissions, these sections were combined into one row for analysis. Public IP addresses are recorded by Qualtrics. However, a unique IP address was not required because different users on the same private network (behind the same gateway or VPN service) may have the same public IP address. Log file analysis was not used.

Each survey response was verified to determine that it was from a unique individual, that the demographics section of the survey was completed, and that the demographic responses confirmed the individual was eligible to participate. Among those meeting that verification standard, response demographics were cross-referenced to the electoral roll sample. The following classifications were used: “direct match”, where survey ID and/or demographics fully matched to the electoral roll sample; “uniquely reconciled”, where there were incomplete or typographical errors in the ID or demographics, but a unique match was able to be made to the electoral roll sample; and “non-uniquely reconciled” where a response was able to be verified as from an eligible participant, but a unique match to a specific individual on the electoral roll sample list was not certain (for instance, change of address, or only year of birth provided, etc.).

We estimated that the survey would take between 10 and 15 min to complete due to the extensive adaptive questioning within the survey. Respondents’ time to completion was recorded; an a priori minimum time limitation of five minutes was specified as the threshold for valid responses to be included. The completion rate was calculated based on the unique number of people submitting the last page of the questionnaire divided by the number of people who had unique and valid responses.

#### 2.6.1. Pilot Testing

The first 50 responses were reviewed as pilots. No issues were identified with the survey’s deployment or completion that warranted a change in methodology. Based on the piloting, we updated the survey information to reflect a 10–20 min survey completion time and added additional demographic verification fields in case participants had lost access to the original participant survey ID.

#### 2.6.2. Data Protection

We developed a comprehensive data management plan that we incorporated into the ethical approval process. The plan included processes outlining data collection and storage. Importantly, files containing personal information from the electoral roll or the survey, such as addresses, were stored separately from all other data on a secured network drive, accessible only by the project team. After verification, survey responses were delinked from any personal data and stored only with the survey ID. All survey data were collated into the Qualtrics platform, hosted by the University of Auckland, and then transferred for storage and analysis to a secure network drive, administered by NZ e-Science Infrastructure (NeSI), and held for 6 years [23].

### 2.7. Analysis

#### 2.7.1. Missing Data

After survey validation checks, all unique responses that completed demographics required for ID verification and a single question of self-identified disability identity were included. No data were imputed for any questionnaire responses.

#### 2.7.2. Māori Iwi and Rohe Data

The questionnaire provided space for respondents to specify up to five iwi (tribes) and associated rohe (tribal areas) per the iwi statistical standard. This Stats NZ standard provides guidelines for how to gather, organise, and report iwi and iwi-related groups’ information and statistics. This standard is useful to those who collect iwi information, including iwi, Māori, government agencies, and researchers. Assigning iwi classification codes was completed by a single Māori investigator in Stata 17 (StataCorp LLC, College Station, TX, USA) and involved initially reconciling and reformatting entries for consistency of spelling and, especially, the use of macrons. These free-text variables were then mapped to the Stats NZ “Iwi and iwi-related groups statistical classification V1.0.0” [24], and grouped into 18 categories: 12 categories group iwi by geographical regions; a further 4 categories describe situations where there is incomplete information (confederations and waka, iwi not named; Iwi named, region not known; hapū affiliated to more than one iwi; and region known, iwi not named); and 2 final categories capture responses that are completely unidentifiable, or missing.

#### 2.7.3. Statistical Correction—Weighting

Survey weights were applied to account for any variations between our respondents and the Māori-descent population of NZ. This was undertaken in two steps. First, we weighted our respondents to the random sample drawn from the electoral roll Māori descent population. Second, because the electoral roll systematically misses some people from the full population, we weighted the electoral roll Māori descent population to the estimated adult Māori descent population for June 2022, based on the Administrative Population Census (APC) maintained by Stats NZ [25].

Weighting respondents to the electoral roll: We used a logistic regression model to weight our respondents to the electoral roll, with response (yes/no) as the dependent variable and the following demographic characteristics as predictors: gender (male, female), age (18–29, 30–44, 45–54, 55–64, 65–74, 75+), region (Northland, Auckland, Waikato, Bay of Plenty, Hawke’s Bay/Gisborne, Taranaki/Manawatū-Whanganui, Wellington, Tasman/Nelson/Marlborough/West Coast, Canterbury, Otago/Southland), quintiles of deprivation using the NZDep2018 deprivation score [26], and occupation mapped to the “major group” occupations of the Australian and New Zealand Classification of Occupations (ANZSCO) codes (managers, professionals, technicians and trades workers, community and personal service workers, clerical and sales workers, machinery operators, drivers and labourers, students, retirees, others not in labour force, not stated) [27]. The model included all main effects and any significant two-way interactions. Weights are taken as the inverse of predicted probabilities, standardised so that the mean weight = 1.

Weighting the electoral roll to the Administrative Population Census Māori descent population: The New Zealand Electoral Commission (2013) reports that 93% of the Māori descent estimated resident population aged 18 years and over is enrolled on either the Māori descent or general electoral roll. Aggregate electoral roll enrolment data may conceal over- and under-coverage compared with the estimated resident population, meaning that those of Māori descent on the electoral roll are not a random sample of those of Māori descent in the population [28]. For example, electoral roll coverage is significantly lower for people aged between 18 and 25 years. Therefore, we chose to weight the electoral roll to the Māori descent population from the APC. Stats NZ maintains and makes available an ongoing count of the estimated population based on administrative records of the APC [25,28]. The APC derives census-type information from linked administrative data in the Integrated Data Infrastructure (IDI) [25]. We obtained these data for the adult (aged 18+ years) Māori descent population as of June 2022 (n = 526,476), stratified by 10 regions, 4 age groups, 5 deprivation quintiles, and 2 genders (400 strata). A weight was derived for each of these strata by calculating the APC strata proportion divided by the electoral roll strata proportion.

A final population weight was derived by multiplying the electoral roll weight by the APC weight, relativising to 1, and then multiplying by the inverse of the overall population sampling fraction (i.e., the inverse of the achieved sample, n = 7230, divided by the overall adult Māori descent population, n = 526,476). This weight allows us to estimate the number of Māori in NZ who have our outcomes of interest.

## 3. Results

### 3.1. Sample Size and Eligible Participants

Figure 2 shows a flow diagram of the number of participants who were eligible, enrolled, and included in the dataset for analysis. From the combined electoral rolls (N = 527,598), a random sample was drawn of 70,155 (13.3%) people of Māori descent, consisting of 36,212 from the Māori roll and 33,943 from the general roll. From the drawn sample of 70,155 people, 3947 individuals had addresses that were invalid or undeliverable. In addition, we received notification on behalf of another 33 individuals, informing us that they were unable to complete the survey because they did not meet the eligibility criteria or were deceased.

Our first tranche of survey invitations consisted of 40,157 letter packs, which were sent in July 2022, and by the end of August, these had resulted in 3974 responses across the three response mediums (online, phone, and mail options), as shown in Table 1. Consequently, the second tranche of 29,998 letter packs was sent in September 2022, as originally planned. Table 1 details the sample sizes, mailing dates, and number of responses for both tranches. In total, 66,175 people received the questionnaire and were eligible participants.

The survey remained open from July to December 2022, as shown in Figure 3. Two postcard reminders were sent following the initial invitation letter, as survey reminders are recognised as an important component of survey deployment [29], and these also proved successful for this survey.

### 3.2. Participation Rate

From the 66,175 eligible participants, a total of 7359 responses were received, giving a response rate of 11.1%. The median completion time was 22 min, and the shortest response was 6 min—longer than the a priori minimum threshold of 5 min completion; consequently, nobody was excluded. The majority, 6820 (92.7%) of participants, elected to complete the survey online. In our survey, 7.3% of participants opted for alternate methods of survey completion: 398 (5.4%) by interviewer-administered phone option, with a further 141 (2.0%) completing a paper survey. Participants who opted for alternate methods of survey completion were overrepresented among adults 65 years and older (21.1%), retirees (26.2%), those not in the labour force (17.2%), those self-identifying as disabled (17.1%), and those living in the highest deprivation (11.1%) (see Table 2).

Our verification check of the IDs of the 7359 participants revealed there were 7230 who completed the demographics section. Twenty-nine participants (0.4%) who provided insufficient demographic information for us to confirm their identity or eligibility were therefore excluded. Of those 7230 participants, 7126 (98.6%) were a direct match, and 51 were uniquely reconcilable, leaving 53 who were non-uniquely reconcilable to our electoral roll sample. However, those 53 were all confirmed as unique individuals who each met the eligibility criteria, and therefore all were included in the analysis. Therefore, a total of 7230 participants were included in the analysis.

### 3.3. Estimation of Sampling Error

The sampling error was calculated per the sampling design: a simple random sample with post-stratification weights applied to account for non-response and under-coverage of the sampling frame. We calculated our actual sample margin of error as 1.1% [30].

### 3.4. Completion Rate

In total, 6774 (93.7%) participants answered the last question, “life satisfaction”. Among those who responded online, the completion rate was 93.4%. Three hundred and eighty-two (96.2%) participants who elected to respond by phone completed the survey, and of the participants who requested a paper copy of the survey, 138 (97.9%) responses were complete. Of the 153 items, 145 (94.7%) had a completion rate greater than 90%, and no item had less than an 84% completion rate—see Appendix A (Figure A1). The 8 items with the lowest completion rate were personal income and context related to discrimination; these had completion rates ranging between 84.5% and 89.5%. The discrimination items with low response rates were validated items from the Te Kupenga survey, but instead of individual “yes/no” responses to each item, the response patterns appear to indicate that the items have been frequently misinterpreted as “tick all that apply”.

### 3.5. Sample Representativeness & Statistical Weighting

Of the 527,598 people of Māori descent in both Māori and General rolls, 9689 records were removed as they were non-New Zealand residents. From the resulting 517,909 people in the electoral roll extract, a simple random sample was drawn as the sample framework for recruitment in this study.

### 3.6. Weighting to Electoral Roll

Comparisons of our participants to the electoral roll sample for the variables of gender, age group, region, occupation, NZDep2018 quintile, and urbanicity—see Appendix A (Table A3). Females were relatively overrepresented compared to the electoral roll sample, as were older age categories. There were also geographical variations, with underrepresentation in more northern regions (Northland, Auckland, Bay of Plenty, and Gisborne) and overrepresentation in more southern regions (Wellington, Canterbury, and Otago). Participants tended to be distributed to lower NZDep2018 deprivation. Occupations that were overrepresented were managers, professionals, clerical and sales workers, and retirees. Some demographic characteristics resulted in low cell counts (i.e., older age groups, provincial regions, and related occupations) and were combined before creating models for weighting. The binary logistic regression model constructed to predict the probability of responding showed a high level of significance (*p* < 0.001) for all variables except urbanicity (*p* = 0.140). Of the interactions explored, five were found to be significant: Gender x age group, gender x deprivation, gender x occupation, age group x occupation, and region by deprivation—see Appendix A (Table A4). The final model consisted of five main effects and five significant two-way interactions—see Appendix A (Table A5).

Appendix A Figure A2 and Figure A3 show graphical representations of the participant demographics and results of the sample weighting to the electoral roll. Figure A2 compares the demographics of the participants with the sample framework (i.e., the electoral roll extract) to compare differences before the weighting was applied. This comparison was performed to assess response bias between those participants who were invited and those who completed the survey. Based on this analysis, a weighting model was applied to adjust the survey responses to be representative of the electoral roll Māori descent population. Figure A3 demonstrates that the weighting was able to fully and effectively rectify all of the aforementioned variations, and the weighted sample is fully representative of the electoral roll population.

### 3.7. Weighting to the Administrative Population Census

Appendix A Figure A4 and Figure A5 shows the same comparison with weighting to the APC. Figure A4 compares the electoral roll-weighted characteristics of participants with the APC. This represents a comparison to the full New Zealand population of people of Māori descent. We found the participant weighting from the electoral roll sample demonstrated a relative overrepresentation amongst females and a significant under-representation amongst 18- and 29-year-olds, compared with the APC population. The regional comparison suggested that the electoral weighted sample was overrepresentative for Auckland, the Bay of Plenty, and Wellington and underrepresentative for Canterbury. NZDep2018 comparisons were similar for both but with overrepresentation in the higher deprivation quintile. Figure A5 shows the effect of adding the second weight multiplier to further adjust the participant characteristics back to the full APC. This weighting was able to achieve a very accurate representation across all weighted variables.

## 4. Discussion

This is the first large-scale, independent, representative survey of Māori, designed specifically to understand the prevalence and impact of disability in NZ; an area that is widely recognised as underprioritized [5]. We have presented the methodology and design of a survey that resulted in over 7200 participants of Māori descent. It is also the first survey to frame disability in a health and wellbeing context by purposefully including social and cultural constructs of te ao Māori. Importantly, the development was undertaken using a KMR approach, with a multi-disciplinary and community-participatory team, of which the majority were Māori, had lived experience of disability, or both.

The large sample population achieved by our survey with voluntary participation is comparable with sample sizes achieved by NZ government-administered and mandated surveys [31,32]. For example, Te Kupenga 2013 had 5500 participants, and 2018 had 8500 participants of Māori ethnicity or descent [32]. Te Kupenga provides key statistics on four areas of Māori cultural wellbeing: wairuatanga (spirituality), tikanga (Māori customs and practices), te reo Māori (the Māori language), and whanaungatanga (social connectedness). It does not, however, include measures of health or wellbeing. In contrast, the NZ Health Survey (2021/2022) reports the health and wellbeing of 4434 adult participants, of whom only 803 were of Māori ethnicity. Whereas our survey has a Māori adult population almost an order of magnitude larger, meaning it is well-placed to provide more meaningful and representative interpretations of Māori health and wellbeing.

### 4.1. Sample Framework

Large NZ government-run surveys use complex sampling frameworks, but these are not publicly available and usually involve door-to-door area sampling, amongst other methods, for improving response rates. Furthermore, many of these government surveys are compulsory as legislated within the Statistics Act 2022, and non-compliance is considered an offence. The only feasible framework from which we could draw our sample was the electoral roll. This provides a nationwide representative list of people who identify as being of Māori descent [28]. The advantages of this sample frame are the high coverage and the fact that it includes some demographic information but limited contact details. It was considered unfeasible to use iwi data as our framework, as these are not centralised, may contain duplicates, i.e., an individual registered with multiple iwi, and do not include people of Māori descent who are not registered with their Iwi [28,33,34].

There are, however, some limitations to using the electoral roll. The electoral roll is typically updated by citizens in the lead-up to an election and has the possibility of inaccuracy, including out-of-date addresses and people who have since died, and it is acknowledged to have low enrolment amongst Māori young people [34]. Although demographic information is collected, only limited information is available for research, such as name, age, physical address, occupation, and title/honorific. This results in a limited means of contacting people and the ability to validate respondents. While it would have been ideal to have a sample frame that identifies disabled populations, there is no administrative data set available that enables direct contact with this specific population for research purposes. Even if such a data set existed, it would be limited to those who identified as disabled using specified criteria and would not have permitted a nuanced exploration of disability from a spectrum of health and wellbeing or from a holistic indigenous perspective. The unavailability of adequate sample frames is an issue that has been identified in other Indigenous populations [8].

Our original intent was to upload and data-link the survey results with the Integrated Data Infrastructure (IDI), a national platform of de-identified administrative data sets operated by Stats NZ [25,33]. However, this posed two key logistical issues. Firstly, ethical requirements of data linkage for “future unspecified research” required participants to agree to a substantive amount of data sovereignty compliance and information, which would have created additional participant burden and potentially biased our results by negatively impacting the response rate. Secondly, there are significant backlogs in Stats NZ’s ability to upload the new data sets, which would have meant several years of delay and an inability to validate the survey sample in a timely manner [33]. Finally, several studies have identified ethical and privacy concerns, particularly amongst young people and Māori, about data misuse with secondary use of data and fear of re-identification in the IDI [35,36]. In contrast, an advantage of our survey being fully self-contained is that it minimises numerator–denominator bias because data are collected in the same context [37].

### 4.2. Maximising Response Rate

We achieved a response rate of 11.1%. Past research has explored reasons for survey non-response [38], and these include factors at both the macro level (society, culture, and economic situation), the meso level (survey design), and the micro level (the respondent). From a theoretical perspective, influences on all levels have been shown to be important [38,39]. We were conscious that even with this knowledge, survey response rates and completion rates have both continued to decline in NZ over the last 30 years [39,40]. To ensure both high response and completion rates, we paid special attention to those factors we could control around survey design and cultural acceptability.

Surveys that originate from the hegemonies of the Global North and are often framed in reductionist paradigms and include deficit-based constructs [38,41] do not get a high response rate among Māori and other Indigenous peoples [40]. We attempted to pre-empt this issue by undertaking a literature search for health, wellbeing, and disability question sets and then identifying those sets that have been created by and validated in Indigenous populations. However, the literature is sparse in this area, so we took a twin track approach. We opted to use existing national and international surveys to maintain comparability and use this as an opportunity to validate these in a Māori population. In doing so, we recognised that these “standardised” sets do not necessarily align with Indigenous Māori concepts of health, wellbeing, and disability. Additionally, there has been no opportunity to form a collective, post-colonial understanding of Māori disability identity. Consequently, we needed to methodologically underpin the existing survey concepts of disability with a holistic, culturally acceptable, wellbeing framework derived from a robust KMR approach. As part of this process, we undertook extensive and iterative qualitative engagement with tāngata whaikaha Māori in the preliminary phase of the survey’s development. This process informed us of the relevant constructs with which to underpin the multidimensional concepts of health, wellbeing, and disability. It also resulted in the need to construct new items, for example, disability kupu (words), societal inclusion, structural discrimination, and ableism. We believe that we have undertaken a novel and culturally respectful means of selecting and creating new items that incorporate the diverse realities of Māori across the continuum of wellbeing.

Given the constraints of using validated questions from predominantly non-indigenous sources, the design elements (the look and feel of the survey) were particularly important. The integration of the cultural elements throughout the survey journey, from invitation to completion, contributed to building cultural credibility. Additionally, the grounding of the survey constructs in a te ao Māori paradigm aimed to minimise self-identification bias arising from potential discordance of worldviews resulting from the framing and language used in the standardised national and international surveys. Related to establishing cultural credibility is the recognised association between response rates and the level of trust that indigenous people have in the team administering the survey, their interpretation of the data, and ethical concerns around data sovereignty [8,35,37]. We suggest that the constitution of our research team, which is predominantly Māori and has lived experience of disability, added to its credibility and helped mitigate these concerns. In keeping with KMR approaches, we ensured that all our information sheets and the online survey welcome page profiled our cultural connections along with the strengths-based, tikanga approach we used for the survey, design, and analysis.

### 4.3. Flexibility: Survey Modes and Formats

Online surveys have increased in popularity for a variety of reasons, including low cost, quick access to data, and non-contact delivery, especially since 2020 and the COVID-19 pandemic [42]. A further advantage is the flexibility that allows participants to complete the survey whenever and wherever they prefer, with their choice of device or platform [43]. However, online surveys have limitations for those with poor access or low familiarity with technology. It has been demonstrated in older populations that not offering a paper survey as an alternative will exclude a small but important group, potentially biasing the results [43]. Our data showed that over seven percent of participants opted for alternate survey modes (phone or paper surveys), demonstrating that this was an important contribution to ensuring the inclusion of older adults, people not in the labour force, those living in deprivation, and people self-identifying as disabled. Likewise, Digital New Zealand analysis confirms that Māori, Pacific peoples, provincial populations, the unemployed, and disabled populations are also less likely to have access to the internet [44]. Their data further shows that those who experience the digital divide report lower wellbeing outcomes than those with digital access, making this an important confounder for health and wellbeing-related surveys. Without the use of alternate modes in our survey, we would have systematically excluded health and wellbeing data from several intersectionally disadvantaged groups and biased our results.

It has been suggested that hybrid survey designs incorporating web-based and other completion modes lead to quicker and higher response rates [9,45,46]. Despite their logistical advantages, hybrid designs can lead to compromises in the validity of data if methodological care is not taken [46,47]. Combining alternate collection modes can result in variations in coverage, nonresponse rates, and measurement errors, which can reduce the comparability of the data. De Leeuw reported that mixing modes necessitates “an explicit trade off of errors and costs” (p. 235) [47]. In our study, we were able to avoid the self-selection bias and other impacts of these issues by ensuring that all participants, irrespective of mode, were drawn from a systematic sample framework, were presented with identical items in each mode, and used experienced telephone interviewers who used a standardised approach for the phone data collection [46,47].

As well as the alternate modes, making the survey formats as universally accessible as possible was a key consideration for the team. The need for specific accessible formats such as Braille, large print, and EasyRead is required for a small proportion of the population, but there is no administrative data set that could proactively identify these potential participants. We were, therefore, unable to deploy appropriate accessible formats to the relevant people in our sample frame. This may mean that we have potentially excluded some people who required specific accessible formats to access, interpret, and understand the mailed survey invitation. The Electoral Commission Disability Strategy 2020, created in 2013 [48], is committed to improving access to the Electoral Commission’s services and information in ways that meet their needs. Despite this, there remains no mechanism to identify who on the electoral roll might require alternative access needs. Consequently, we focused on using plain language in the information sheet and consent form and providing a variety of methods to contact the team for further in-person support.

Other contributing strategies that we implemented to maximise the response rate were the inclusion of two postcard reminders and a prize draw. Reminders are recognised as an important component of survey deployment [29], and in our experience, these proved very successful. Interestingly, the second reminder in both tranches proved almost as beneficial as the first reminder. We also offered all participants the opportunity to go into a prize draw. We were mindful of respondent burden and conscious of survey length. Had we been able to link our data set to the IDI, we would have been able to limit our questions because this information would have been able to be collected from linked administrative data sets. Consequently, the survey length and mean time to complete were longer than we would have preferred, but our extensive use of adaptive questioning aimed to mitigate this issue. While attribution of any one of these specific factors is not possible, we are confident that the combination of all these measures was successful, as evidenced by the extremely high completion rate (93.7%), demonstrating that, despite its length, the survey was culturally engaging and acceptable.

### 4.4. Weighting and Generalisability

We included robust processes to ensure our sample was generalisable to the Māori-descent population. The results of our weighting to the electoral roll sample demonstrated that participants were significantly different in several demographic variables [49]. Prior to weighting, there were response biases in favour of older people, females, several geographical regions, and lower deprivation. Similarly, under-coverage was found for men and young people in the 2018 Te Kupenga survey and other household surveys, but not for geographical regions or deprivation. The first weighting process was able to fully adjust for all observed biases, ensuring that our data were fully generalisable to the electoral roll population.

A novel methodological aspect of the study was the secondary weighting of our sample according to the newly developed APC. The APC is part of Stats NZ’s census transformation programme, which aims to use longitudinal administrative NZ resident population data to replace cross-sectional census data. These data have only been available since 2021, and this latest APC provides access for the purposes of weighting to a reference population otherwise unavailable for sampling purposes.

This secondary weighting process highlighted the relative underrepresentativeness of young people in the electoral roll, as previously described [28]. It additionally highlighted differences between the electoral roll and the APC for gender and deprivation, with the electoral roll underrepresenting males and having the highest deprivation. It is possible that some of these variations may be a result of a random sampling error that may have occurred from the simple random sample of the electoral roll. However, given the sample size of over 70,000 (13% of the total electoral roll), those differences are likely to be minor. Despite this, the secondary weighting to the APC allowed us to weight the entire population on all demographic variables, something not possible in the electoral roll itself because of the necessary demographic imputation. Reassuringly, our extensive quality assurance has demonstrated that our weighted survey data are of high quality and that weighted estimates are in line with expectations for all other demographics [50].

## 5. Conclusions

Our research has identified and implemented approaches to survey design and deployment that are by, with, and for tāngata whaikaha Māori. We have demonstrated a high level of engagement and participation by Māori in this survey. Our experience highlights the significant extent to which Indigenous codesign was essential to ensuring that quantitative survey methods were accessible and culturally appropriate for Māori. We have highlighted several areas where structural deficits in system-level knowledge (the lack of holistic Indigenous models of wellbeing in national datasets and health and disability surveys) and the failure to prioritise an accessibility lens in national data infrastructure, including the electoral roll and IDI, continue to compromise Indigenous research approaches. These areas require additional development to ensure high-quality data and information that are responsive to the epistemological aspirations of tāngata whaikaha Māori.

## Figures and Tables

**Figure 1 ijerph-20-06797-f001:**
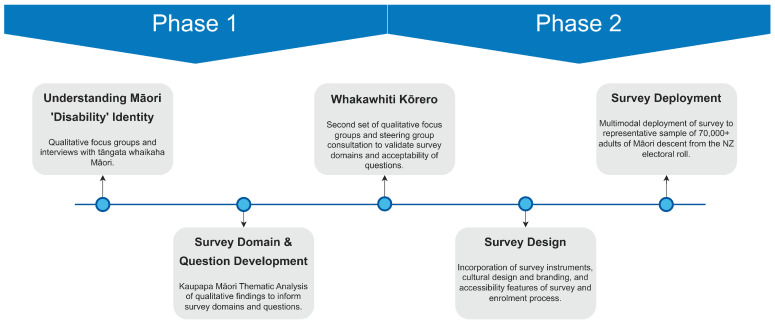
Conceptual overview showing main methodological components to survey development and deployment.

**Figure 2 ijerph-20-06797-f002:**
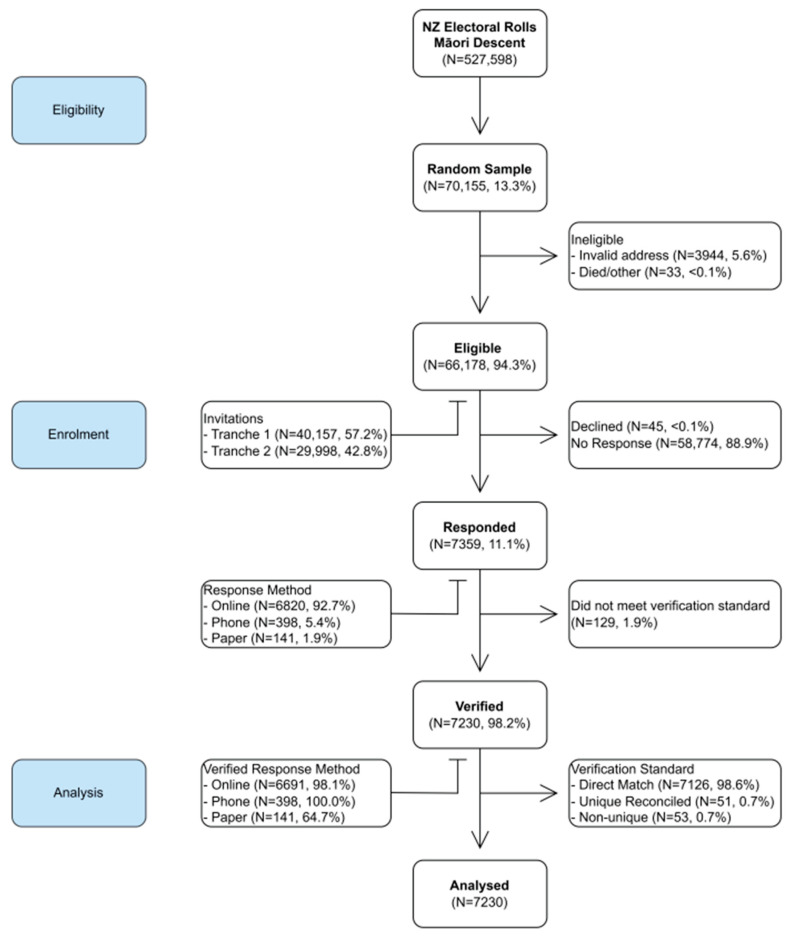
Flow diagram of participant progress through phases of the study eligibility and enrolment processes.

**Figure 3 ijerph-20-06797-f003:**
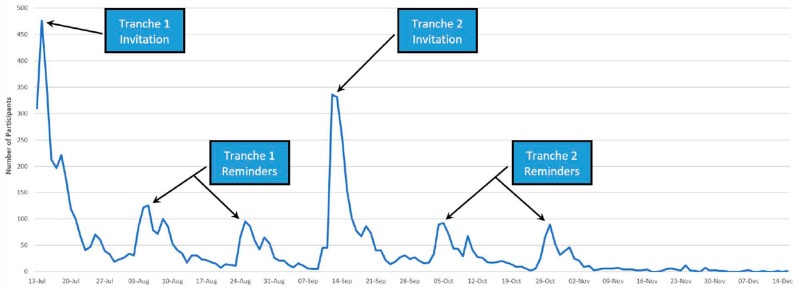
Recruitment progress showing responses to invitations and reminders.

**Table 1 ijerph-20-06797-t001:** Invitation tranches, reminders, and response rates.

Invitation Tranche	Date	Number (Crude Response Rate %)
First sample invitation letters	12 July 2022	40,157
First sample reminder postcard 1	2 August 2022	36,664
First sample reminder postcard 2	23 August 2022	35,065
First sample paper questionnaires	Various, as requested	98
**First sample total responses**	**By 16 December 2022**	**4311 (10.7%)**
Second sample invitation letters	8 September 2022	29,998
Second sample reminder postcard 1	29 September 2022	27,269
Second sample reminder postcard 2	20 October 2022	24,222
Second sample paper questionnaires	Various, as requested	119
**Second sample total responses**	**By 16 December 2022**	**2866 (9.6%)**

**Table 2 ijerph-20-06797-t002:** Responses and participant demographics by mode of completion.

	Online Responses (% of Responses)	Phone Responses (% of Responses)	Paper Responses (% of Responses)	All Alternate Modes n (% Total)
**Gender**				
Male	2589 (38.7%)	185 (46.5%)	69 (48.9%)	254 (8.9%)
Female	4102 (61.3%)	213 (53.5%)	72 (51.1%)	285 (6.5%)
**Age group**				
18–29	941 (14.1%)	7 (1.8%)	-	7 (0.7%)
30–44	1610 (24.1%)	21 (5.3%)	3 (2.1%)	24 (1.5%)
45–54	1489 (22.3%)	39 (9.8%)	3 (2.1%)	42 (2.7%)
55–64	1399 (20.9%)	107 (26.9%)	25 (17.7%)	132 (8.6%)
65–74	931 (13.9%)	130 (32.7%)	49 (34.8%)	179 (16.1%)
75+	319 (4.8%)	94 (23.6%)	61 (43.3%)	155 (32.7%)
**Region**				
Northland	477 (7.1%)	28 (7.0%)	13 (9.2%)	41 (7.9%)
Auckland	1557 (23.3%)	88 (22.1%)	32 (22.7%)	120 (7.2%)
Waikato	843 (12.6%)	50 (12.6%)	19 (13.5%)	69 (7.6%)
Bay of Plenty	696 (10.4%)	52 (13.1%)	12 (8.5%)	64 (8.4%)
Gisborne	157 (2.3%)	15 (3.8%)	5 (3.5%)	20 (11.3%)
Hawke’s Bay	347 (5.2%)	23 (5.8%)	9 (6.4%)	32 (8.4%)
Taranaki	190 (2.8%)	16 (4.0%)	3 (2.1%)	19 (9.1%)
Manawatū-Whanganui	442 (6.6%)	29 (7.3%)	13 (9.2%)	42 (8.7%)
Wellington	813 (12.2%)	37 (9.3%)	12 (8.5%)	49 (5.7%)
Tasman	60 (0.9%)	2 (0.5%)	1 (0.7%)	3 (4.8%)
Nelson	56 (0.8%)	1 (0.3%)	-	1 (1.8%)
Marlborough	58 (0.9%)	6 (1.5%)	1 (0.7%)	7 (10.8%)
West Coast	41 (0.6%)	2 (0.5%)	2 (1.4%)	4 (8.9%)
Canterbury	577 (8.6%)	29 (7.3%)	10 (7.1%)	39 (6.3%)
Otago	244 (3.6%)	12 (3.0%)	5 (3.5%)	17 (6.5%)
Southland	133 (2.0%)	8 (2.0%)	4 (2.8%)	12 (8.3%)
**Urbanicity**				
Major urban	3989 (59.8%)	224 (57.1%)	70 (49.6%)	294 (6.9%)
Minor urban	1468 (22.0%)	104 (26.5%)	34 (24.1%)	138 (8.6%)
Rural	1209 (18.1%)	64 (16.3%)	37 (26.2%)	101 (7.7%)
**NZDep2018**				
Lowest deprivation	1010 (15.2%)	23 (5.9%)	13 (9.2%)	36 (3.4%)
Second quintile	1053 (15.8%)	39 (9.9%)	16 (11.3%)	55 (5.0%)
Third quintile	1164 (17.5%)	45 (11.5%)	25 (17.7%)	70 (5.7%)
Fourth quintile	1492 (22.4%)	93 (23.7%)	36 (25.5%)	129 (8.0%)
Highest deprivation	1947 (29.2%)	192 (49.0%)	51 (36.2%)	243 (11.1%)
**Occupation**				
Managers	700 (10.5%)	18 (4.6%)	5 (3.5%)	23 (3.2%)
Professionals	1334 (20.1%)	30 (7.7%)	7 (5.0%)	37 (2.7%)
Technicians and trades workers	448 (6.7%)	20 (5.1%)	12 (8.5%)	32 (6.7%)
Community and personal service workers	425 (6.4%)	30 (7.7%)	6 (4.3%)	36 (7.8%)
Clerical workers	626 (9.4%)	16 (4.1%)	3 (2.1%)	19 (2.9%)
Sales workers	204 (3.1%)	3 (0.8%)	1 (0.7%)	4 (1.9%)
Machinery operators and drivers	212 (3.2%)	16 (4.1%)	5 (3.5%)	21 (9.0%)
Labourers	384 (5.8%)	24 (6.2%)	9 (6.4%)	33 (7.9%)
Students	469 (7.1%)	7 (1.8%)	2 (1.4%)	9 (1.9%)
Retirees	442 (6.7%)	101 (25.9%)	56 (39.7%)	157 (26.2%)
Others Not in labour force	617 (9.3%)	101 (25.9%)	27 (19.1%)	128 (17.2%)
Not stated	785 (11.8%)	24 (6.2%)	8 (5.7%)	32 (3.9%)
**Self-identified** **disability**				
Yes	1281 (19.2%)	206 (52.0%)	59 (44.0%)	265 (17.1%)
No	5399 (80.8%)	190 (48.0%)	75 (56.0%)	265 (4.7%)

## Data Availability

Data are contained within the article and Appendix A only.

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
