# Peer review of "Measuring Māori Health, Wellbeing, and Disability in Aotearoa Using a Web-Based Survey Methodology"

_ijerph, 2023, doi:10.3390/ijerph20186797_

Round 1
Reviewer 1 Report
I found this to be an exemplary manuscript in terms of detail and quality of writing. The survey is a very important piece of work for several reasons, and the processes involved in its development, administration and deriving results, are carefully outlined here. As such, there were only minor formatting details that I've picked up for the authors to attend to:
Page 2 line 52 – delete the word and
Line 56, delete the word and between disability and multidimensional poverty?
Line 69 – taonga doesn’t have a macron
Page 3, line 111 – add comma after ‘best practice,’
Lines 123-127 – delete as these are duplication
Page 5, line 189 – of surveys that can dissuade responses
Line 230 – insert comma after steering group
Line 233 – insert comma after online
Line 244 – identified as being of Māori descent?
Lines 450-452 – delete these instructions
Lines 577-8 – identify as being of Māori descent
Glossary – alternative definition for iwi? The term tribe/s has colonial connotations
Author Response
We thank the reviewer sincerely for their positive review and useful suggestions. Responses below, IN ALL CAPS FOR CALRITY
-
Page 2 line 52 – delete the word and - REMOVED
-
Line 56, delete the word and between disability and multidimensional poverty? - REMOVED
-
Line 69 – taonga doesn’t have a macron - REMOVED
-
Page 3, line 111 – add comma after ‘best practice,’ - ADDED
-
Lines 123-127 – delete as these are duplication - REMOVED
-
Page 5, line 189 – of surveys that can dissuade responses - CORRECTED
-
Line 230 – insert comma after steering group - ADDED
-
Line 233 – insert comma after online - ADDED
-
Line 244 – identified as being of Māori descent? - ADDED
-
Lines 450-452 – delete these instructions - REMOVED [OOPS!,]
Lines 577-8 – identify as being of Māori descent - ADDED
Glossary – alternative definition for iwi? The term tribe/s has colonial connotations - REDEFINED THIS TERM ALONG WITH SEVEN OTHER TERMS TO PROVIDE MORE ROBUST DEFINITIONS PER YOUR RECOMMENDATION
Reviewer 2 Report
Introduction:
· After stating research problem, give strong justification why this study needs to be done
· Acknowledge the work of other researchers on the topic and identify gaps in knowledge which need to be filled
· Clearly describe General and Specific objectives of the study
Methodology:
· This section is weak, needs improvement
· Describe briefly the study design
· When the study was conducted
· Describe the study population and sampling frame
· Describe recruitment process including inclusion and exclusion criteria
· Describe criteria for selection of 15 member steering group
· Qualitative methods were used to inform understanding of Māori disability identity, explain which qualitative methods were used , such as interviews of Key Informants ? Focus Group Discussions ?
Study participants were invited through personalized letter, what was the refusal rate ? Or proportion of invited people didn’t respond in spite of reminders
Results:
· Summarize your results according to the objectives of the study
· Total 66,175 people were eligible, however only 7,359 ( 11,1%) responded , which is very low response rate, it might have introduced bias in the study
Please explain
Discussion:
Summarize the important findings of your study and compare results of your study with similar published studies
Conclusion:
Restate your research problem, summarize your main findings, highlight the significance of your findings
no comments
Author Response
We thank the reviewer for their comments. We politely note, however, that from our perspective, all of the points raised in the review are addressed in detail in the manuscript itself. We provide reference to the line numbers where these issues are addressed, point by point - IN ALL CAPS FOR CLARITY.
Introduction:
- After stating research problem, give strong justification why this study needs to be done - PROBLEM STATEMENT Lines 40-71, JUSTIFICATION FOR STUDY NEED Lines 72-81, DESCRIPTION OF APPROACH Lines 82-83.
- Acknowledge the work of other researchers on the topic and identify gaps in knowledge which need to be filled - EXISTING & GAPS Lines 58-71. THESE ARE FURTHER REFERENCED IN THE DISCUSSION.
- Clearly describe General and Specific objectives of the study - GENERAL OBJECTIVE Lines 82-83, SPECIFIC OBJECTIVES Lines 115-122,
Methodology:
- This section is weak, needs improvement - THIS SECTION CONTAINS ALL ELEMENTS OF THE CHERRIES GUIDELINES AND IS PRESENTED IN DEVELOPMENTALLY CHRONOLOGICAL ORDER. REFERENCE TO SECTION AND PAGE NUMBER OF CHERRIES CRITERIA ARE SUPPLIED.
- Describe briefly the study design - REFERENCE TO THIS BEING A NATIONALLY REPRESENTIVE SURVEY - Line 87-88, Lines 115-118
- When the study was conducted - Line 464
- Describe the study population and sampling frame - Lines 243-263
- Describe recruitment process including inclusion and exclusion criteria - Lines 265 - 283
- Describe criteria for selection of 15 member steering group - FOR CLARITY, THE RESEARCHERS DID NOT SELECT GROUP MEMBERS, WE PARTNERED WITH AN EXISTING NATIONAL GROUP AND THEIR LOCAL SUB-GROUP - Line 106, and ACKNOWLEDGED AT lines 766-767. WE HAVE AMENDED THE TEXT TO MMAKE THIS CLEARER.
- Qualitative methods were used to inform understanding of Māori disability identity, explain which qualitative methods were used , such as interviews of Key Informants ? Focus Group Discussions ? - THESE METHODS ARE PUBLISHED IN DETAIL ELSEWHERE - Lines 140-141 Reference 5. THE METHODS INCLUDED INTERVIEWS AND FOCUS GROUPS TO COLLECT DATA, WE USED WĀNANGA FOR VALIDATION - Lines 150-158.
Study participants were invited through personalized letter, what was the refusal rate ? Or proportion of invited people didn’t respond in spite of reminders - PARTICIPATION RATE - Lines 472-473 & Table 1
Results:
- Summarize your results according to the objectives of the study - THIS IS A METHODOLOGY PAPER. THE RESULTS RELATE TO THE PROCESS OF CONDUCTING THE SURVEY AS OUTLINED BY CHERRIES GUIDELINES.
- Total 66,175 people were eligible, however only 7,359 ( 11,1%) responded , which is very low response rate, it might have introduced bias in the study. Please explain. - THE BIASES ARE DETAILED IN SECTIONS 3.6 & 3.7, TABLES 1-3, AND FIGURES 1-4. THESE ARE DISCUSSED AT LENGTH 713-740
Discussion:
Summarize the important findings of your study and compare results of your study with similar published studies. - IMPORTANT FINDINGS Lines 553-560, COMPARISONS Lines 561-571
Conclusion:
Restate your research problem, summarize your main findings, highlight the significance of your findings - PROBLEM Lines 742-743 & 746-751. FINDINGS Lines 743-744. SIGNIFICANCE Lines 744-746.
Reviewer 3 Report
This paper demonstrates the extreme importance of ensuring that researchers can and do gather accurate and reliable data that can be used to benefit Māori communities. It details the extent to which a survey tool to measure health, wellbeing, and disability must be redesigned and effectively transformed when surveying Māori. From the outset, the design of the survey tool was determined by Māori, using well-tested Māori methodological approaches which draw on Māori experience, expertise and frameworks so that the tool would relevant to and therefore useful for Māori. The response rate to the survey demonstrated that Māori, unsurprisingly, will engage more productively with such a survey and that the data obtained will be far more reliable.
The manner in which the design and deployment of the survey tool uses western survey design considerations and statistical methods but reconfigures every aspect of them to cater for the lived realities of Māori is impressive. These are considerations that are make significant differences to the uptake and success of a survey involving Māori and it is clear that this research team has been able to understand and correctly draw on and apply the expertise of knowledge holders within the Māori community.
Transforming a nationally representative health, wellbeing and disabilities survey to make it relevant to Māori at the level of detail undertaken by this team, and then achieving a survey of 7,230 participants was a massive undertaking that is also fundamentally important to providing better data for Māori and hence the possibility of better outcomes. It is also directly and immediately relevant to other fields of research that rely on accurate data about Māori and other Indigenous communities. Ka nui aku mihi ki a koutou i tēnei mahi rangatira.
Author Response
We wish to thank the reviewer for their detailed review and supportive comments. There were no issues raised that require amendment to the manuscript.
Tenei te mihi atu ki a koe e te rangatira mō tō awhi mai i a mātou
(thank you for your support of our work)